# Structures of Mammeasins P and Q, Coumarin-Related Polysubstituted Benzofurans, from the Thai Medicinal Plant *Mammea siamensis* (Miq.) T. Anders.: Anti-Proliferative Activity of Coumarin Constituents against Human Prostate Carcinoma Cell Line LNCaP

**DOI:** 10.3390/ph16020231

**Published:** 2023-02-03

**Authors:** Fenglin Luo, Yoshiaki Manse, Saowanee Chaipech, Yutana Pongpiriyadacha, Osamu Muraoka, Toshio Morikawa

**Affiliations:** 1Pharmaceutical Research and Technology Institute, Kindai University, 3-4-1 Kowakae, Higashi-osaka 577-8502, Osaka, Japan; 2Faculty of Agro-Industry, Rajamangala University of Technology Srivijaya, Thungyai, Nakhon Si Thammarat 80240, Thailand; 3Faculty of Science and Technology, Rajamangala University of Technology Srivijaya, Thungyai, Nakhon Si Thammarat 80240, Thailand

**Keywords:** *Mammea siamensis*, mammeasin, polysubstituted coumarin, coumarin-related polysubstituted benzofuran, anti-proliferative activity, prostate carcinoma, Calophyllaceae

## Abstract

A methanol extract of the flowers of *Mammea siamensis* (Miq.) T. Anders. (Calophyllaceae) showed anti-proliferative activity against human prostate carcinoma LNCaP cells (IC_50_ = 2.0 µg/mL). Two new coumarin-related polysubstituted benzofurans, mammeasins P (**1**) and Q (**2**), and a known polysubstituted coumarin mammea B/AC cyclo F (**39**) were isolated from the extract along with 44 previously reported polysubstituted coumarin constituents (**3**–**38** and **40**–**47**). The structures of two new compounds (**1** and **2**) were determined based on their spectroscopic properties derived from the physicochemical evidence including NMR and MS analyses and taking the plausible generative pathway into account. Among the coumarin constituents, mammeasins A (**3**, IC_50_ = 1.2 µM) and B (**4**, 0.63 µM), sugangin B (**18**, 1.5 µM), kayeassamins E (**24**, 3.0 µM) and G (**26**, 3.5 µM), and mammeas E/BA (**40**, 0.88 µM), E/BB (**41**, 0.52 µM), and E/BC (**42**, 0.12 µM) showed relatively potent anti-proliferative activity.

## 1. Introduction

Prostate cancer, a hormonally driven cancer, is the second most frequent malignancy in men worldwide [1,2,3,4]. It may be asymptomatic at an early stage, often has an indolent course, and may require minimal or even no treatment. However, the most frequent complaints are difficulty with urination, increased urination frequency, and nocturia, all of which may also arise from prostatic hypertrophy. A more advanced stage of the disease may present with urinary retention and back pain, as the axial skeleton is the most common site of bone metastasis [4]. The main therapeutic option for advanced prostate cancer is androgen deprivation therapy, which has limited clinical outcomes. However, its therapeutic benefit does not last long, and most patients develop metastatic castration-resistant prostate cancer. After progression into castration-resistant prostate cancer, several chemotherapeutic drugs are used. Chemotherapy is the standard first-line treatment for recurrent metastatic castration-resistant prostate cancer, and relapse eventually occurs due to drug resistance [5]. Therefore, there is a strong demand for the development of new therapeutic molecules against prostate cancer.

The Calophyllaceae plant *Mammea siamensis* (Miq.) T. Anders., which is the name recorded by the World Flora Online [6], is a small evergreen tree with fragrant yellow or white flowers distributed throughout Thailand (locally known as Sarapi or Saraphi), Laos, Cambodia, Vietnam, and Myanmar [7,8,9,10,11]. In Thailand, the flower part of this medicinal plant has been used as a heart tonic for lowering fever, and for boosting appetite [12,13]. Previously, several coumarin [9,10,11,14,15,16,17,18,19,20,21,22,23,24], xanthone [8,10,17,21], flavonoid [7,23], triterpene [10,12], and steroid [10,12,24] constituents have been reported from the flowers [7,9,10,11,12,13,14,15,16,20,23,24], seeds [18,21], twigs [8,17,19], and bark [22] of *M. siamensis*. This plant and its constituents have been reported to possess anti-proliferative and apoptotic activities [9,13,22,23,24], B-cell-specific Moloney murine leukemia virus insertion region 1 promoter inhibitory activity [11], and anti-malarial activity [10]. In our study, the coumarin constituents obtained from the flowers of *M. siamensis* were reported to have suppressive effects on inducible nitric oxide synthase expression in RAW264.7 cells [25], aromatase [26,27], and 5*α*-reductase [28] inhibitory activities as well as anti-proliferative activities against human digestive tract carcinoma cells, such as human gastric carcinoma HSC-2, HSC-4, and MKN-45 cells [29]. Coumarins have also attracted much attention for their anti-proliferative activity against prostate cancer cells, such as the human prostate carcinoma cell line LNCaP [30,31,32]. Further studies on the flowers of *M. siamensis* revealed that a methanol extract exhibited anti-proliferative activity against LNCaP cells. Thus, this study deals with further separation studies on the chemical constituents of this extract for the isolation and structure determination of two new compounds, mammeasins P (**1**) and Q (**2**), as well as the identification of active coumarin constituents with anti-proliferative activity against LNCaP cells.

## 2. Results and Discussion

### 2.1. Anti-Proliferative Effects of the Methanol Extract and Its Fractions against Human Prostate Carcinoma LNCaP Cells

LNCaP, one of the most commonly used cell lines for prostate cancer research, is derived from a human lymph node metastatic lesion in prostate adenocarcinoma [33,34,35]. These cells are androgen-responsive because they show mRNA and protein expression of the androgen receptor and prostate-specific antigen [36,37]. As shown in Table 1, the anti-proliferative effect of a methanol extract of the dried flowers of *M. siamensis* (25.66% dried material) against human prostate carcinoma LNCaP cells was observed (IC_50_ = 2.0 µg/mL). Bioassay-guided fractionation of the methanol extract with ethyl acetate (EtOAc)-H_2_O (1:1, *v*/*v*) yielded an EtOAc-soluble fraction (6.84%) and aqueous phase. The latter was subjected to Diaion HP-20 column chromatography (H_2_O→MeOH) according to previously reported protocols, which yielded H_2_O- and MeOH-eluted fractions (13.50% and 4.22%, respectively) [25]. The EtOAc-soluble fraction showed the highest activity (IC_50_ = 2.7 µg/mL), whereas the other fractions showed no noticeable activity.

### 2.2. Isolation

From the active EtOAc-soluble fraction, we isolated 45 known coumarin constituents (**3**–**38** and **40**–**47**), including the newly obtained compound mammea B/AC cyclo F (**39**, 0.0005%) [15], using normal-phase silica gel and reversed-phase ODS column chromatographic purification and finally HPLC [27,28,29]. In this study, mammeasins P (**1**, 0.0004%) and Q (**2**, 0.0005%) were newly isolated (Figure 1).

### 2.3. Structure Determination for Mammeasins P (***1***) and Q (***2***)

Mammeasin P (**1**) was obtained as a pale-yellow oil, and its molecular formula was determined to be C_22_H_26_O_6_ via high-resolution electron ionization (EI)-MS at *m*/*z* 386.1724 (M^+^, calcd for 386.1729). The ^1^H- and ^13^C-NMR spectra of **1** (Table 2, CDCl_3_) were assigned with the aid of distortionless enhancement by polarization transfer (DEPT), ^1^H–^1^H correlation spectroscopy (COSY), heteronuclear single-quantum coherence (HSQC), and heteronuclear multiple-bond connectivity (HMBC) (Figure 2). The ^1^H-NMR spectrum showed signals for five methyl moieties [*δ* 1.01 (3H, t, *J* = 7.4 Hz, H_3_-4’’’), 1.27 (3H, t, *J* = 7.6 Hz, H_3_-3’), 1.52 (6H, s, 2’’-CH_3_ × 2), and 3.71 (3H, s, 2-COOCH_3_)], four methylene moieties [*δ* 1.73 (2H, qt, *J* = 7.4, 7.4 Hz, H_2_-3’’’), 2.67 (2H, q, *J* = 7.6 Hz, H_2_-2’), 3.07 (2H, t, *J* = 7.4 Hz, H_2_-2’’’), and 3.75 (2H, s, H_2_-3)], a pair of *cis*-substituted olefinic protons [*δ* 5.51 and 6.65 (1H each, both d, *J* = 9.7 Hz, H-3’’ and H-4’’)], and a hydrogen-bonded hydroxy proton [*δ* 14.12 (1H, s, 8a-OH)]. ^1^H−^1^H COSY experiments on **1** indicated the presence of partial structures (bold lines in Figure 2). In HMBC experiments, long-range correlations were observed between the following proton and carbon pairs: H_2_-3 and C-2, 4, 4a, 1’; H_2_-2’ and C-4, 1’; H-3’’ and C-6, 2’’, *C*H_3_-2’’; H-4’’ and C-5, 7, 2’’; C*H*_3_-2’’ and C-2’’, 3’’; H_2_-2’’’, H_2_-3’’’ and C-1’’’; 8a-OH and C-4a, 8, 8a. Thus, a polysubstituted benzofuran skeleton in **1** was constructed, and the linkage positions of the *n*-butyryl and 2,2-dimethyl-2*H*-pyran groups in **1** were clarified. This benzofuran skeleton is speculated to be derived from 4-(1’-acetoxypropyl)coumarin, a common structure of many mammeacoumarins, via intramolecular displacements to the furocoumarins, acid-catalyzed double-bond migration, and lactone opening (Figure 3) [38,39]. Thus, the structure of **1** was determined.

The molecular formula of mammeasin Q (**2**) was determined to be C_23_H_28_O_6_, showing a molecular ion peak at *m/z* 400.1880 (M^+^, calcd for 400.1886), using high-resolution EI-MS measurements. The ^1^H- and ^13^C-NMR spectra (Table 1, CDCl_3_) of **2** were similar to those of **1**, except for the signals owing to a 2-methyl-butyryl moiety in C-8 position [*δ* 0.92 (3H, t, *J* = 7.5 Hz, H_3_-4’’), 1.17 (3H, d, *J* = 6.9 Hz, H_3_-5’’’), 1.42, 1.87 (1H each, both m, H_2_-3’’’), and 3.81 (1H, m, H-2’’’)] instead of an *n*-butyryl moiety, as seen in **1**. As shown in Figure 2, the connectivity of the quaternary carbons in **2** was elucidated via ^1^H–^1^H COSY and HMBC experiments. ^1^H−^1^H COSY correlations indicated the presence of the following partial structures of **2**, shown in bold lines: linkage of C-2’–C-3’; C-3‘–C-4’’; C-2’’’–C-5’’’. HMBC correlations revealed long-range correlations between the following proton and carbon pairs: H_2_-3 [*δ* 3.75 (2H, s)] and C-2, 4, 4a, 1’; H_2_-2’ [*δ* 2.67 (2H, q, *J* = 7.5 Hz)] and C-4, 1’; H-3’’ [*δ* 5.51 (1H, d, *J* = 9.8 Hz)] and C-6, 2’’, *C*H_3_-2’’; H-4’’ [*δ* 6.65 (1H, d, *J* = 9.8 Hz)] and C-5, 7, 2’’; C*H*_3_-2’’ [*δ* 1.52 (6H, s)] and C-2’’, 3’’; 8a-OH [*δ* 14.09 (1H, s)] and C-4a, 8, 8a; H_2_-2’’’, H_2_-3’’’, H_3_-5’’’ and C-1’’’. Therefore, the structure of **2** was established.

### 2.4. Anti-Proliferative Effects of the Coumarin Constituents against Human Prostate Carcinoma LNCaP Cells

Chemical studies on the flowers of *M. siamensis* allowed the isolation of the two above-mentioned new coumarin-related polysubstituted benzofurans, mammeasins P (**1**) and Q (**2**), and 45 coumarin constituents (**3**–**47**) (Figure 4). As has been previously reported [25,26,27,28,29], these isolates were obtained from the active EtOAc-soluble fraction of the methanol extract to furnish the following isolation yields from the dried material: mammeasins A (**3**, 0.0250%), B (**4**, 0.0083%), C (**5**, 0.0008%), D (**6**, 0.0047%), E (**7**, 0.0102%), F (**8**, 0.0015%), G (**9**, 0.0025%), H (**10**, 0.0009%), I (**11**, 0.0008%), J (**12**, 0.0006%), K (**13**, 0.0008%), L (**14**, 0.0006%), M (**15**, 0.0021%), N (**16**, 0.0007%), and O (**17**, 0.0015%); surangins B (**18**, 0.0272%), C (**19**, 0.0571%), and D (**20**, 0.0600%); kayeassamin A (**21**, 0.0578%), 8-hydroxy-5-methyl-7-(3,7-dimethylocta-2,6-dienyl)-9-(2-methyl-1-oxobutyl)-4,5-dihydropyrano [4,3,2-*de*]chromen-2-one (**22**, 0.0015%), 8-hydroxy-5-methyl-7-(3,7-dimethylocta-2,6-dienyl)-9-(3-methyl-1-oxobutyl)-4,5-dihydropyrano [4,3,2-*de*]chromen-2-one (**23**, 0.0012%); kayeassamins E (**24**, 0.0125%), F (**25**, 0.0435%), G (**26**, 0.0196%), and I (**27**, 0.0107%); mammeas A/AA (**28**, 0.0528%), A/AB (**29**, 0.0054%), A/AC (**30**, 0.1055%), A/AD [=mesuol (**31**, 0.0036%)], A/AA cyclo D (**32**, 0.0039%), A/AB cyclo D (**33**, 0.0363%), A/AC cyclo D (**34**, 0.0109%), A/AA cyclo F (**35**, 0.0010%), A/AC cyclo F (**36**, 0.0119%), B/AB cyclo D (**37**, 0.0016%), B/AC cyclo D (**38**, 0.0062%), B/AC cyclo F (**39**, 0.0005%), E/BA (**40**, 0.0045%), E/BB (**41**, 0.0288%), E/BC (**42**, 0.0130%), E/BC cyclo D (**43**, 0.0058%), and E/BD cyclo D (**44**, 0.0007%); and deacetylmammeas E/AA cyclo D (**45**, 0.0025%), E/BB cyclo D (**46**, 0.0056%), and E/BC cyclo D (**47**, 0.0078%). To characterize the active constituents, the anti-proliferative effects of the isolates against LNCaP cells were examined. However, evaluation of the coumarin constituents (**21**–**23**, **29**, **31**, **32**, and **39**), including the new compounds **1** and **2**, for which we did not have a sufficient sample amount to assess biological activity, could not be performed. As shown in Table 3, seven coumarin constituents, including mammeasins A (**3**, IC_50_ = 1.2 µM) and B (**4**, 0.63 µM), sugangin B (**18**, 1.5 µM), kayeassamins E (**24**, 3.0 µM) and G (**26**, 3.5 µM), and mammeas E/BA (**40**, 0.88 µM), E/BB (**41**, 0.52 µM), and E/BC (**42**, 0.12 µM), showed relatively potent anti-proliferative activity. The structural requirements of the coumarins for the activity were suggested as the following: (1) regardless of the structure of the substitutions at C-4, C-5, C-6, and C-8, coumarins with 7-OH group (e.g., **3**, **4**, **18**, **24**, **26**, **40**, **41**, and **42**) were essential for the strong activity; (2) compounds with the 6-prenylcoumarin moieties showed stronger or equivalent activity to those with the geranyl moiety [**41** > **18**, **42** > **3**, **40** ≒ **4**] or those forming the 2,2-dimethyl-2*H*-pyran structure with 5-OH group [**41** > mammea E/BD cyclo D (**44**, IC_50_ = 53.9 µM), **42** > mammea E/BC cyclo D (**43**, 23.1 µM)]; (3) compounds with the 4-propylcoumarin moieties with the 1’-acetoxy group showed stronger activity than those of the corresponding deacetylcoumarins [**4** > surangin D (**20**, 24.7 µM), **18** > surangin C (**19**, 11.8 µM)] or those forming the 2-methyl-3,4-dihydro-2*H*-pyran structure with 5-OH group [**3** > mammeasin D (**6**, 25.0 µM), **4** > mammeasin C (**5**, 30.5 µM)].

## 3. Materials and Methods

### 3.1. Spectroscopy and Column Chromatography

The following instruments were used to obtain spectroscopic data: specific rotation, JASCO P-2200 polarimeter (JASCO Corporation, Tokyo, Japan, *l * =  5 cm); UV spectra, Shimadzu UV-1600 spectrometer; IR spectra, IRAffinity-1 spectrophotometer (Shimadzu, Kyoto, Japan); ^1^H NMR spectra, JNM-ECA800 (800 MHz), JNM-LA500 (500 MHz), JNM-ECS400 (400 MHz), and JNM-AL400 (400 MHz) spectrometers; ^13^C NMR spectra, JNM-ECA800 (200 MHz), JNM-LA500 (125 MHz), JNM-ECA400 (100 MHz), and JNM-AL400 (100 MHz) spectrometers (JEOL, Tokyo, Japan); EI-MS and high-resolution EI-MS, JMS-GCMATE mass spectrometer (JEOL, Tokyo, Japan); HPLC detector, SPD-10A*vp* UV-VIS detector; and HPLC columns, Cosmosil 5C_18_-MS-II (Nacalai Tesque, Kyoto, Japan). For NMR, the samples were dissolved in deuterated chloroform (CDCl_3_) at room temperature with tetramethylsilane as an internal standard. Columns of 4.6 mm × 250 mm and 20 mm × 250 mm were used for analytical and preparative purposes, respectively.

The following chromatographic materials were used for column chromatography (CC): highly porous synthetic resin, Diaion HP-20 (Mitsubishi Chemical, Tokyo, Japan); normal-phase silica gel CC, silica gel 60 N (Kanto Chemical, Tokyo, Japan; 63–210 mesh, spherical, neutral); reversed-phase ODS CC, Chromatorex ODS DM1020T (Fuji Silysia Chemical, Aichi, Japan; 100–200 mesh); TLC, pre-coated TLC plates with silica gel 60F_254_ (Merck, Darmstadt, Germany, 0.25 mm) (normal-phase) and silica gel RP-18 WF_254S_ (Merck, 0.25 mm) (reversed-phase); and reversed-phase HPTLC, pre-coated TLC plates with silica gel RP-18 WF_254S_ (Merck, 0.25 mm). Detection was performed by spraying with 1% Ce(SO_4_)_2_–10% aqueous H_2_SO_4_, followed by heating.

### 3.2. Plant Material

*M. siamensis* flowers were collected from Nakhonsithammarat Province, Thailand, in September 2006, as previously described [25,26,27,28,29]. The plant material was identified by one of the authors (Y.P.). A voucher specimen (2006.09. Raj-04) was deposited in our laboratory.

### 3.3. Extraction and Isolation

The methanolic extract (25.66% dried material) of the dried flowers of *M. siamensis* (1.8 kg) was partitioned using a solution of EtOAc-H_2_O (1:1, *v*/*v*) to yield an EtOAc-soluble fraction (6.84%) and aqueous phase. The EtOAc-soluble fraction (89.45 g) was subjected to normal-phase silica gel column chromatography [3.0 kg, *n*-hexane-EtOAc (10:1→7:1→5:1, *v*/*v*)→EtOAc→MeOH] to produce 11 fractions [Fr. 1 (3.05 g), Fr. 2 (2.86 g), Fr. 3 (11.71 g), Fr. 4 (1.62 g), Fr. 5 (4.15 g), Fr. 6 (6.29 g), Fr. 7 (2.21 g), Fr. 8 (2.94 g), Fr. 9 (10.23 g), Fr. 10 (11.17 g), and Fr. 11 (21.35 g)], as previously reported [25]. Fraction 2 (2.86 g) was subjected to reversed-phase silica gel CC [74 g, MeOH–H_2_O (70:30→90:10, *v*/*v*)→MeOH→acetone] to yield nine fractions [Fr. 2-1 (21.0 mg), Fr. 2-2 (26.2 mg), Fr. 2-3 (114.1 mg), Fr. 2-4 (425.0 mg), Fr. 2-5 (182.8 mg), Fr. 2-6 (79.6 mg), Fr. 2-7 (94.8 mg), Fr. 2-8 (1211.4 mg), and Fr. 2-9 (328.8 mg)], as described previously [27]. Fraction 2-3 (114.1 mg) was purified via HPLC [Cosmosil 5C_18_-MS-II, MeOH–1% aqueous AcOH (80:20, *v*/*v*)] to give mammeasins P (**1**, 4.8 mg, 0.0004%) and Q (**2**, 6.7 mg, 0.0005%). Fraction 5 (4.15 g) was subjected to reversed-phase silica gel CC [120 g, MeOH–H_2_O (80:20→85:15, *v*/*v*)→MeOH→acetone] to obtain six fractions [Fr. 5-1 (115.7 mg), Fr. 5-2 (2789.8 mg), Fr. 5-3 (515.4 mg), Fr. 5-4 (430.0 mg), Fr. 5-5 (119.2 mg), and Fr. 5-6 (1110.0 mg)], as previously reported [28]. Fraction 5-2 (517.0 mg) was purified via HPLC [Cosmosil 5C_18_-MS-II, MeOH–1% aqueous AcOH (85:15, *v*/*v*) or MeOH–1% aqueous AcOH (80:20, *v*/*v*)] to give mammea B/AC cyclo F (**39**, 1.2 mg, 0.0005%) together with mammeasins M (**15**, 5.0 mg, 0.0021%) and O (**17**, 3.7 mg, 0.0015%) and mammeas A/AA (**28**, 101.2 mg, 0.0418%), A/AC (**30**, 112.9 mg, 0.0466%), A/AA cyclo D (**32**, 3.7 mg, 0.0015%), A/AC cyclo F (**36**, 4.6 mg, 0.0019%), E/BC cyclo D (**43**, 14.0 mg, 0.0058%), and E/BD cyclo D (**44**, 1.8 mg, 0.0007%) [27,28,29].

#### 3.3.1. Mammeasin P (**1**)

Pale-yellow oil; UV [MeOH, nm (log *ε*)]: 247 (4.23), 267 (4.03), 354 (3.29); IR (film): 1743, 1622, 1456, 1167, 1120; ^1^H-NMR (500 MHz, CDCl_3_) *δ*: see Table 1; ^13^C-NMR data (125 MHz, CDCl_3_) *δ*_C_: see Table 1; 2D-NMR spectra: see Appendix A; EIMS *m*/*z* (%): 386 (M^+^, 71), 371 (100); high-resolution EIMS *m*/*z* 386.1724 (calculated for C_22_H_26_O_6_, 386.1729).

#### 3.3.2. Mammeasin Q (**2**)

Pale-yellow oil; [α]^22^_D_ 0 (*c =* 0.34, CHCl_3_); UV [MeOH, nm (log *ε*)]: 247 (4.23), 268 (4.04), 352 (3.36); IR (film): 1743, 1622, 1456, 1169, 1123 cm^–1^; ^1^H-NMR (500 MHz, CDCl_3_) *δ*: see Table 1; ^13^C-NMR data (125 MHz, CDCl_3_) *δ*_C_: see Table 1; 2D-NMR spectra: see Appendix A; EIMS *m*/*z* (%): 400 (M^+^, 100); high-resolution EIMS *m*/*z* 400.1880 (calculated for C_23_H_28_O_6_, 400.1886).

### 3.4. Bioassay

#### 3.4.1. Reagents

RPMI 1640 medium was purchased from FUJIFILM Wako Pure Chemical Industries (Osaka, Japan), fetal bovine serum (FBS) from Biosera (Nuaille, France), other chemicals from FUJIFILM Wako Pure Chemical Industries (Osaka, Japan), and 96-well microplates from Sumitomo Bakelite (Tokyo, Japan).

#### 3.4.2. Cell Culture Assay

Experiments were performed in accordance with previously reported methods [40,41], with slight modifications. LNCaP clone FGC (89110211) was purchased from KAC (Kyoto, Japan). Cells were cultured in RPMI 1640 medium (FUJIFILM Wako) supplemented with 10% FBS, 1 mM sodium pyruvate, 100 U/mL penicillin G, and 100 µg/mL streptomycin at 37 °C in a 5% CO_2_ environment. LNCaP cells were seeded in 96-well plates at a density of 5 × 10^3^ cells/well in 100 µL/well medium, and 100 μL/well of medium containing a test sample was added after an initial incubation of 24 h. Cell viability was detected after 96 h of incubation using the Cell Counting Kit-8 (CCK-8). CCK-8 was purchased from Dojindo Molecular Technologies (Kumamoto, Japan). The O.D. of the yellow-colored formazan solution was measured using a microplate reader at 450 nm (reference: 650 nm) (Appendix A). The IC_50_ value was determined graphically, and the inhibition (%) was calculated using the following formula:Inhibition (%) = [(O.D. (sample) − O.D. (control))/(O.D. (normal) − O.D. (control))] × 100

Each test compound was dissolved in DMSO and added to the medium (final concentration in 0.1% DMSO).

#### 3.4.3. Statistical Analysis

Values are expressed as the mean ± standard error (S.E.M.). One-way analysis of variance (ANOVA) followed by Dunnett’s test was used for statistical analysis. Probability (*p*) values less than 0.05 were considered significant.

## 4. Conclusions

The methanol extract of the flowers of *M. siamensis* (Miq.) showed anti-proliferative activity against human prostate carcinoma LNCaP cells (IC_50_ = 2.0 µg/mL). Two new coumarin-related polysubstituted benzofurans, mammeasins P (**1**) and Q (**2**), were isolated, and their structures were elucidated based on their spectroscopic properties derived from the physicochemical evidence including NMR and MS analyses as well as considering the plausible generative pathway. We have already achieved the total syntheses of mammeasins C (**5**) and D (**6**), which were isolated as new compounds from the same plant material [42], and we would like to conduct the similar synthetic studies for **1** and **2** to further confirm the stated properties in the future.

Among the isolates, seven coumarin constituents, including mammeasins A (**3**, IC_50_ = 1.2 µM) and B (**4**, 0.63 µM), sugangin B (**18**, 1.5 µM), kayeassamins E (**24**, 3.0 µM) and G (**26**, 3.5 µM), and mammeas E/BA (**40**, 0.88 µM), E/BB (**41**, 0.52 µM), and E/BC (**42**, 0.12 µM), showed relatively potent anti-proliferative activity. The results suggest the following structural requirements of the coumarins: (1) 7-OH group was essential for the strong activity; (2) the 6-prenylcoumarin moieties showed stronger or equivalent activity to those with the geranyl moiety or those forming the 2,2-dimethyl-2*H*-pyran structure with 5-OH group; (3) the 4-propylcoumarin moieties with the 1’-acetoxy group showed stronger activity than those of the corresponding deacetyl analogs or those forming the 2-methyl-3,4-dihydro-2*H*-pyran structure with 5-OH group. We previously reported that several of these coumarins, which exhibit potent anti-proliferative activity against LNCaP cells, have moderate enzymatic inhibitory activity against testosterone 5α-reductase [**3** (IC_50_ = 19.0 µM), **4** (24.0 µM), **24** (33.8 µM), **26** (17.7 µM), **40** (16.2 µM), **41** (16.8 µM), and finasteride (0.12 µM), a commercially available 5α-reductase inhibitor] (Appendix A) [27]. The role of 5α-reductase inhibitors in prostate cancer chemoprevention remains controversial [43,44,45]. Tindall and Rittmaster reported that the inhibition of 5α-reductase represents a valid target for prostate cancer risk reduction and treatment strategy [43]. In contrast, Chau and Figg reported that cancer prevention trials with 5α-reductase inhibitors have shown a decreased incidence of low-grade prostate cancer but a potentially increased risk of high-grade disease [44]. Furthermore, a large population-based prospective study on the risk of prostate cancer in men treated with 5α-reductase inhibitors was conducted by Wallerstedt et al. They concluded that treatment with 5α-reductase inhibitors for lower urinary tract symptoms is safe with respect to prostate cancer risk [45]. Therefore, further studies are needed to elucidate whether these coumarins, which have potent anti-proliferative activity against LNCaP cells with moderate 5α-reductase inhibitory activity, are promising therapeutic candidates for prostate cancer.

## Figures and Tables

**Figure 1 pharmaceuticals-16-00231-f001:**
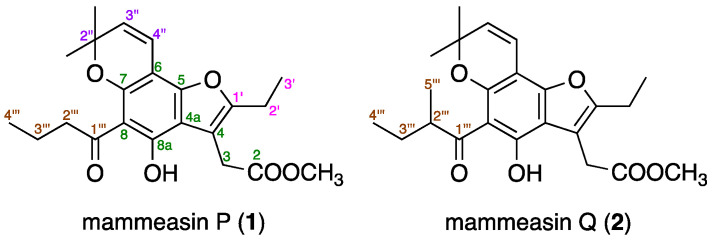
Structures of mammeasins P (**1**) and Q (**2**).

**Figure 2 pharmaceuticals-16-00231-f002:**
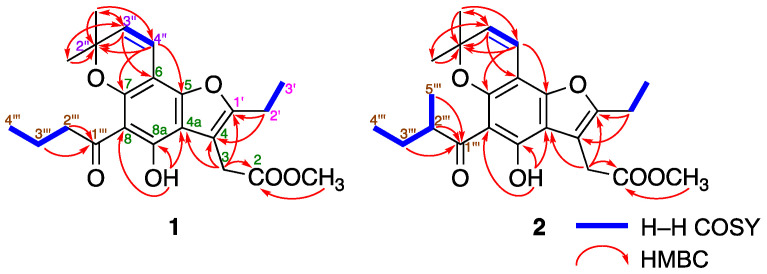
^1^H–^1^H COSY and HMBC correlations of **1** and **2**.

**Figure 3 pharmaceuticals-16-00231-f003:**
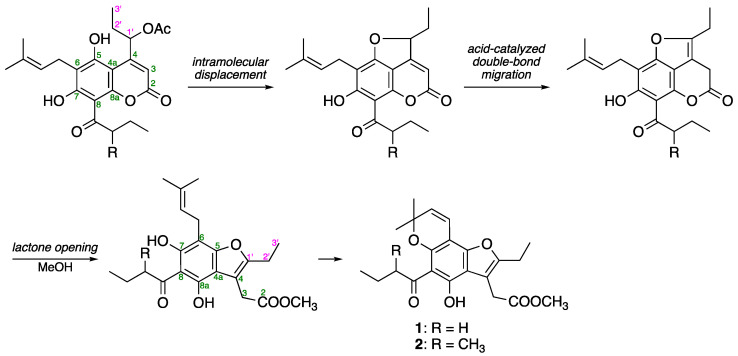
Plausible generative pathway of **1** and **2**.

**Figure 4 pharmaceuticals-16-00231-f004:**
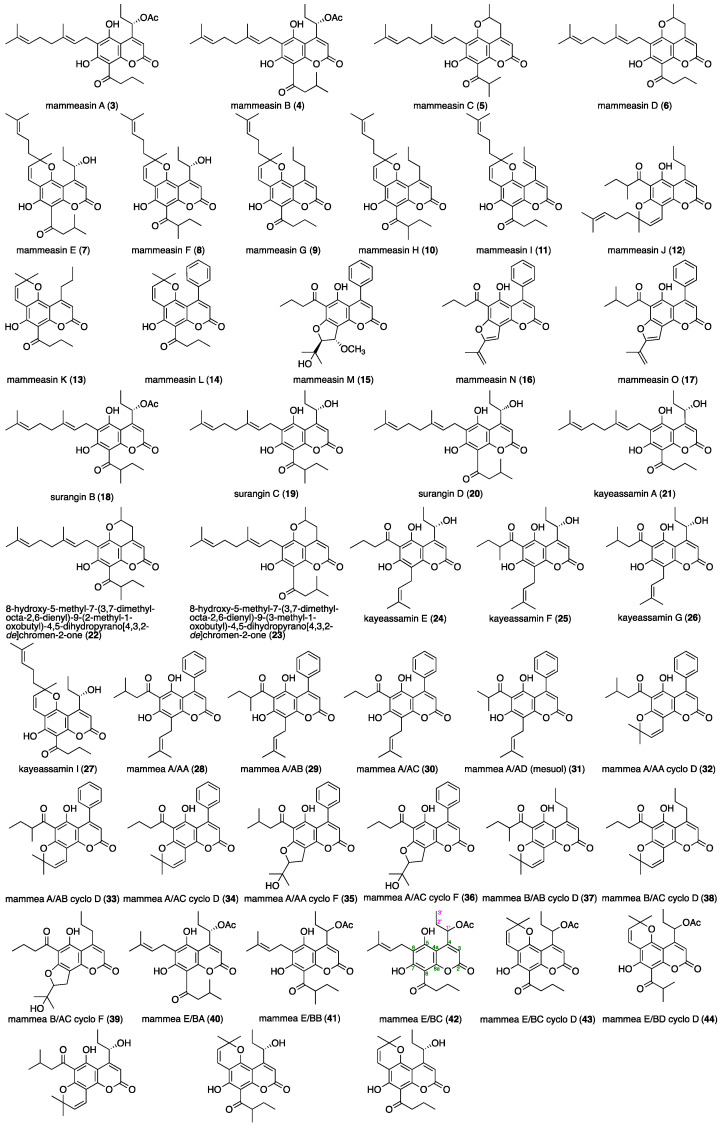
Coumarin constituents (**3**–**47**) from the flowers of *M. siamensis*.

**Table 1 pharmaceuticals-16-00231-t001:** Anti-proliferative effects of the methanol extract of *M. siamensis* flower and its fractions against LNCaP cells.

Treatment	Inhibition (%)					IC_50_
	0 μg/mL	0.3 μg/mL	1 μg/mL	3 μg/mL	10 μg/mL	(μg/mL)
MeOH extract	100.0 ± 1.7	96.2 ± 2.3	87.8 ± 2.0 **	24.5 ± 1.7 **	7.3 ± 0.1 **	2.0
EtOAc-soluble fraction	100.0 ± 1.3	97.7 ± 2.0	97.0 ± 2.3	42.7 ± 1.5 **	7.6 ± 0.2 **	2.7
	**0 μg/mL**	**3 μg/mL**	**10 μg/mL**	**30 μg/mL**	**100 μg/mL**	
MeOH-eluted fraction	100.0 ± 2.0	99.4 ± 4.6	95.8 ± 2.2	38.1 ± 1.5 **	11.9 ± 0.0 **	23.8
H_2_O-eluted fraction	100.0 ± 6.1	99.7 ± 6.3	94.8 ± 4.1	86.1 ± 3.1	72.4 ± 2.5 **	>100

Each value represents the mean ± standard error of the mean (S.E.M.) (*N* = 5). Significantly different from the control (** *p* < 0.01, Dunnett test).

**Table 2 pharmaceuticals-16-00231-t002:** ^1^H and ^13^C NMR spectroscopic data (500 and 125 MHz, CDCl_3_) of mammeasins P (**1**) and Q (**2**).

Position	1		2	
	*δ* _H_	*δ* _C_	*δ* _H_	*δ* _C_
2		171.9		171.9
3	3.75 (2H, s)	29.8	3.75 (2H, s)	29.7
4		107.7		107.7
4a		111.1		111.2
5		* 152.2		* 151.9
6		99.1		99.1
7		* 154.6		* 154.5
8		107.0		107.0
8a		159.8		159.8
2-COO*CH_3_*	3.71 (3H, s)	52.1	3.71 (3H, s)	52.1
8a-OH	14.12 (1H, s)		14.09 (1H, s)	
1’		155.5		155.5
2’	2.67 (2H, q, 7.6)	19.4	2.67 (2H, q, 7.5)	19.5
3’	1.27 (3H, t, 7.6)	12.8	1.27 (3H, t, 7.5)	12.8
2 ’ ’		78.1		78.1
3 ’ ’	5.51 (1H, d, 9.7)	125.7	5.51 (1H, d, 9.8)	125.7
4 ’ ’	6.65 (1H, d, 9.7)	116.1	6.65 (1H, d, 9.8)	116.1
2 ’ ’-CH_3_ × 2	1.52 (6H, s)	27.6	1.52 (6H, s)	27.6
1’’’		207.5		211.8
2’’’	3.07 (2H, t, 7.4)	128.4	3.81 (1H, m)	46.2
3’’’	1.73 (2H, qt, 7.4, 7.4)	18.5	1.42, 1.87 (each 1H, both m)	26.8
4’’’	1.01 (3H, t, 7.4)	14.0	0.92 (3H, t, 7.5)	11.9
5’’’			1.17 (3H, d, 6.9)	16.9

* Assignment may be interchangeable within the same column.

**Table 3 pharmaceuticals-16-00231-t003:** IC_50_ values of anti-proliferative effects of coumarin constituents (**3**–**20**, **24**–**28**, **30**, **33**–**38**, **40**–**47**) from *M. siamensis* flower against LNCaP cells.

Treatment	IC_50_ (µM)	Treatment	IC_50_ (µM)
Mammeasin A (**3**)	1.2	Kayeassamin G (**26**)	3.5
Mammeasin B (**4**)	0.63	Kayeassamin I (**27**)	16.1
Mammeasin C (**5**)	30.5	Mammea A/AA (**28**)	51.9
Mammeasin D (**6**)	25.0	Mammea A/AC (**30**)	26.2
Mammeasin E (**7**)	5.9	Mammea A/AB cyclo D (**33**)	>100 (82.7) ^(^*^a^*^)^
Mammeasin F (**8**)	16.7	Mammea A/AC cyclo D (**34**)	>100 (90.0) ^(^*^a^*^)^
Mammeasin G (**9**)	83.5	Mammea A/AA cyclo F (**35**)	21.3
Mammeasin H (**10**)	69.4	Mammea A/AC cyclo F (**36**)	39.7
Mammeasin I (**11**)	*ca* 100	Mammea B/AB cyclo D (**37**)	61.9
Mammeasin J (**12**)	>100 (86.9) ^(^*^a^*^)^	Mammea B/AC cyclo D (**38**)	>100 (78.4) ^(^*^a^*^)^
Mammeasin K (**13**)	>100 (79.9) ^(^*^a^*^)^	Mammea E/BA (**40**)	0.88
Mammeasin L (**14**)	49.4	Mammea E/BB (**41**)	0.52
Mammeasin M (**15**)	>100 (91.3) ^(^*^a^*^)^	Mammea E/BC (**42**)	0.12
Mammeasin N (**16**)	*ca* 100	Mammea E/BC cyclo D (**43**)	23.1
Mammeasin O (**17**)	35.2	Mammea E/BD cyclo D (**44**)	53.9
Surangin B (**18**)	1.5	Deacetylmammea E/AA cyclo D (**45**)	25.9
Surangin C (**19**)	11.8	Deacetylmammea E/BB cyclo D (**46**)	34.0
Surangin D (**20**)	24.7	Deacetylmammea E/BC cyclo D (**47**)	19.7
Kayeassamin E (**24**)	3.0		**IC_50_ (nM)**
Kayeassamin F (**25**)	6.2	Paclitaxel ^(^*^b^*^)^ [40,41]	3.7

Each value represents the mean ± S.E.M. (*N* = 5). ^(*a*)^ Values in parentheses represent the control cell viability at 100 µM. ^(*b*)^ The positive control, paclitaxel, was purchased from FUJIFILM Wako Pure Chemical Industries (Osaka, Japan) [40,41].

## Data Availability

Data supporting the findings of this study are available from the corresponding author upon reasonable request.

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
