# Peer review of "Structures of Mammeasins P and Q, Coumarin-Related Polysubstituted Benzofurans, from the Thai Medicinal Plant Mammea siamensis (Miq.) T. Anders.: Anti-Proliferative Activity of Coumarin Constituents against Human Prostate Carcinoma Cell Line LNCaP"

_pharmaceuticals, 2023, doi:10.3390/ph16020231_

Round 1
Reviewer 1 Report
1. The introduction part and conclusions are needed to be enlarged with more details.
2. What was the percentage of purity of the isolated molecules?
3. The stated properties of the new molecules must be confirmed with other methods, also.
Author Response
We are grateful to your reviewing our manuscript and providing valuable suggestions to improve the manuscript. We have incorporated all your comments and suggestions in our revised manuscript. I hope this new manuscript is acceptable for publication in Pharmaceuticals.
Reviewer #1
- The introduction part and conclusions are needed to be enlarged with more details.
→ Thank you for your comment. The following additions have been made to the Introduction and Conclusion sections to describe them in more detail.
(p 2 lines 45-60)
The Calophyllaceae plant Mammea siamensis (Miq.) T. Anders., which is recorded to the World Flora Online as an accepted name [6], is a small evergreen tree with fragrant yellow or white flowers distributed throughout Thailand (locally known as Sarapi or Saraphi), Laos, Cambodia, Vietnam, and Myanmar [7–11]. In Thai, the flower part of this medicinal plant has been used as a heart tonic, for lowering fever, and for boosting appetite [12,13]. Previously, several coumarin [9–11,14–24], xanthone [8,10,17,21], flavonoid [7,23], triterpene [10,12], and steroid [10,12,24] constituents have been reported from the flowers [7,9–16,20,23,24], seeds [18,21], twigs [8,17,19], and bark [22] of M. siamensis. This plant and its constituents have been reported to possess anti-proliferative and apoptotic activities [9,13,22–24], B cell-specific Moloney murine leukemia virus insertion region 1 promoter inhibitory activity [11], and anti-malarial activity [10]. In our study, the coumarin constituents obtained from the flowers of M. siamensis were reported to have suppressive effects on inducible nitric oxide synthase expression in RAW264.7 cells [25], aromatase [26,27] and 5a-reductase [28] inhibitory activities as well as anti-proliferative activities against human digestive tract carcinoma cells, such as human gastric carcinoma HSC-2, HSC-4, and MKN-45 cells [29].
(p 9 lines 274-285)
The structural requirements of the coumarins for the activity were suggested as followings; (1) regardless of the structure of the substitutions at C-4, C-5, C-6, and C-8, coumarins having the 7-hydroxy group (e.g. 3, 4, 18, 22, 24, 40, 41, and 42) was essential for the strong activity; (2) compounds having the 6-prenylcoumarin moieties showed stronger or equivalent activity than those having the geranyl moiety [41 > 18, 42 > 3, 40 ≒ 4] or those forms the 2,2-dimethy-2H-pyran structure with the 5-hydroxy group [41 > mammea E/BD cyclo D (44, IC50 = 53.9 µM), 42 > mammea E/BC cyclo D (43, 23.1 µM)]; (3) compounds having the 4-propylcoumarin moieties with the 1'-acetoxy group showed stronger activity than those of the corresponding deacetylcoumarins [4 > surangin D (20, 24.7 µM), 18 > surangin C (19, 11.8 µM)] or those forms the 2-methyl-3,4-dihydro-2H-pyran structure with the 5-hydroxy group [3 > mammeasin D (6, 25.0 µM), 4 > mammeasin C (5, 30.5 µM)].
- What was the percentage of purity of the isolated molecules?
→ Thank you for your comment. As for the purities of all of the isolated molecules were purified grade by preparative HPLC. This is one of the purification method for naturally occurring compounds that are generally commercially available as reagents, and thus corresponds to reagent grade purity. (Sorry, we did not have calculated the percentage of purity of each isolated molecule.) Therefore, we believe that there are no particular problem with the biological findings and structure-activity relationships, which are the main theme and results of this study.
- The stated properties of the new molecules must be confirmed with other methods, also.
→ Thank you for your comment. As for the structural determination of two new compounds (1 and 2), we have determined based on their spectroscopic properties derived from the physicochemical evidence including NMR and MS analyses as well as considering for the plausible generative pathway as shown in Figure 3. The Abstract section has been revised as follows to provide a more detailed description.
(p 1 lines 20–23)
The structures of two new compounds (1 and 2) were determined based on their spectroscopic properties derived from the physicochemical evidence including NMR and MS analyses as well as considering for the plausible generative pathway.
(p 8 lines 263–270)
Two new coumarin-related polysubstituted benzofurans, mammeasins P (1) and Q (2), were isolated, and their structures were elucidated based on their spectroscopic properties derived from the physicochemical evidence including NMR and MS analyses as well as considering for the plausible generative pathway. We have already achieved the total syntheses of mammeasins C (5) and D (6), which are isolated as the new compounds from the same plant material [43], and we would like to conduct the similar synthetic studies for 1 and 2 to further confirm the stated properties in the future.

Reviewer 2 Report
The paper ID 2142896 reports the isolation of the coumarins from Mammea siamensis flowers and thier activity against human prostate carcinoma LNCaP cells. After reviewing, my comments concerning the manuscript are general positive, but before acceptance for publication in Pharmaceuticals, the results need some modification considering following suggestions:
1. In the introduction, please provide more information about the chemical composition of the examined flowers of Mammea siamensis. What is their main profile of action and use in medicine so far? Is it a species known for its use in traditional medicine?
2. The Authors evaluated the in vitro antiproliferative effect of the methanol extract of M. siamensis flower, its fractions and 37 coumarins against LNCaP cells. The extensive activity studies were unfortunately not discussed in the results and discussion section. Apart from listing the activity potential of the 7 most active coumarins, there was no discussion. It seems interesting to compare the activity of individual coumarins to their content in the extract. Therefore, I kindly ask to extend the research with the LC-PDA or LC-PDA-MS analysis to show which compounds dominate in the extract and to what extent they may determine the activity of the extract. I also kindly ask to present a representative LC-PDA chromatogram of the tested extract.
Author Response
We are grateful to your reviewing our manuscript and providing valuable suggestions to improve the manuscript. We have incorporated all your comments and suggestions in our revised manuscript. I hope this new manuscript is acceptable for publication in Pharmaceuticals.
Reviewer #2
The paper ID 2142896 reports the isolation of the coumarins from Mammea siamensis flowers and thier activity against human prostate carcinoma LNCaP cells. After reviewing, my comments concerning the manuscript are general positive, but before acceptance for publication in Pharmaceuticals, the results need some modification considering following suggestions:
- In the introduction, please provide more information about the chemical composition of the examined flowers of Mammea siamensis. What is their main profile of action and use in medicine so far? Is it a species known for its use in traditional medicine?
→ Thank you for your comment. The following additions have been made to the Introduction section to describe them in more detail.
(p 2 lines 45-60)
The Calophyllaceae plant Mammea siamensis (Miq.) T. Anders., which is recorded to the World Flora Online as an accepted name [6], is a small evergreen tree with fragrant yellow or white flowers distributed throughout Thailand (locally known as Sarapi or Saraphi), Laos, Cambodia, Vietnam, and Myanmar [7–11]. In Thai, the flower part of this medicinal plant has been used as a heart tonic, for lowering fever, and for boosting appetite [12,13]. Previously, several coumarin [9–11,14–24], xanthone [8,10,17,21], flavonoid [7,23], triterpene [10,12], and steroid [10,12,24] constituents have been reported from the flowers [7,9–16,20,23,24], seeds [18,21], twigs [8,17,19], and bark [22] of M. siamensis. This plant and its constituents have been reported to possess anti-proliferative and apoptotic activities [9,13,22–24], B cell-specific Moloney murine leukemia virus insertion region 1 promoter inhibitory activity [11], and anti-malarial activity [10]. In our study, the coumarin constituents obtained from the flowers of M. siamensis were reported to have suppressive effects on inducible nitric oxide synthase expression in RAW264.7 cells [25], aromatase [26,27] and 5a-reductase [28] inhibitory activities as well as anti-proliferative activities against human digestive tract carcinoma cells, such as human gastric carcinoma HSC-2, HSC-4, and MKN-45 cells [29].
Reference numbers have been changed to reflect the addition of references due to the above revision
- The Authors evaluated the in vitro antiproliferative effect of the methanol extract of M. siamensis flower, its fractions and 37 coumarins against LNCaP cells. The extensive activity studies were unfortunately not discussed in the results and discussion section.Apart from listing the activity potential of the 7 most active coumarins, there was no discussion. It seems interesting to compare the activity of individual coumarins to their content in the extract. Therefore, I kindly ask to extend the research with the LC-PDA or LC-PDA-MS analysis to show which compounds dominate in the extract and to what extent they may determine the activity of the extract. I also kindly ask to present a representative LC-PDA chromatogram of the tested extract.
→ Thank you for your valuable comment.
- A more detailed discussion of the structural requirements of coumarin for the anti-proliferative effects against LNCaP cells have been added to the Conclusions section.
(p 9 lines 274-285)
The structural requirements of the coumarins for the activity were suggested as followings; (1) regardless of the structure of the substitutions at C-4, C-5, C-6, and C-8, coumarins having the 7-hydroxy group (e.g. 3, 4, 18, 22, 24, 40, 41, and 42) was essential for the strong activity; (2) compounds having the 6-prenylcoumarin moieties showed stronger or equivalent activity than those having the geranyl moiety [41 > 18, 42 > 3, 40 ≒ 4] or those forms the 2,2-dimethy-2H-pyran structure with the 5-hydroxy group [41 > mammea E/BD cyclo D (44, IC50 = 53.9 µM), 42 > mammea E/BC cyclo D (43, 23.1 µM)]; (3) compounds having the 4-propylcoumarin moieties with the 1'-acetoxy group showed stronger activity than those of the corresponding deacetylcoumarins [4 > surangin D (20, 24.7 µM), 18 > surangin C (19, 11.8 µM)] or those forms the 2-methyl-3,4-dihydro-2H-pyran structure with the 5-hydroxy group [3 > mammeasin D (6, 25.0 µM), 4 > mammeasin C (5, 30.5 µM)].
- Isolation yields of each coumarin were added as information from which the contribution of the extracted extract to the activity can be inferred from the content and activity intensity of individual coumarins and renumbered of the compound number as shown in revised Figure 4. We would like to further study for the quantitative determination of the coumarin constituents in siamensis using HPLC or LC-MS will be considered as a future research topic.
(p 6 lines 144-160)
These isolates were obtained from the active EtOAc-soluble fraction of the methanol extract to furnish following isolation yield from the dried material; mammeasins A (3, 0.0250%), B (4, 0.0083%), C (5, 0.0008%), D (6, 0.0047%), E (7, 0.0102%), F (8, 0.0015%), G (9, 0.0025%), H (10, 0.0009%), I (11, 0.0008%), J (12, 0.0006%), K (13, 0.0008%), L (14, 0.0006%), M (15, 0.0021%), N (16, 0.0007%), and O (17, 0.0015%), surangins B (18, 0.0272%), C (19, 0.0571%), and D (20, 0.0600%), kayeassamin A (21, 0.0578%), 8-hydroxy-5-methyl-7-(3,7-dimethylocta-2,6-dienyl)-9-(2-methyl-1-oxobutyl)-4,5-dihydropyrano[4,3,2-de]chromen-2-one (22, 0.0015%), 8-hydroxy-5-methyl-7-(3,7-dimethylocta-2,6-dienyl)-9-(3-methyl-1-oxobutyl)-4,5-dihydropyrano[4,3,2-de]chromen-2-one (23, 0.0012%), kayeassamins E (24, 0.0125%), F (25, 0.0435%), G (26, 0.0196%), and I (27, 0.0107%), mammeas A/AA (28, 0.0528%), A/AB (29, 0.0054%), A/AC (30, 0.1055%), A/AD [= mesuol (31, 0.0036%)], A/AA cyclo D (32, 0.0039%), A/AB cyclo D (33, 0.0363%), A/AC cyclo D (34, 0.0109%), A/AA cyclo F (35, 0.0010%), A/AC cyclo F (36, 0.0119%), B/AB cyclo D (37, 0.0016%), B/AC cyclo D (38, 0.0062%), B/AC cyclo F (39, 0.0005%), E/BA (40, 0.0045%), E/BB (41, 0.0288%), E/BC (42, 0.0130%), E/BC cyclo D (43, 0.0058%), and E/BD cyclo D (44, 0.0007%), and deacetylmammeas E/AA cyclo D (45, 0.0025%), E/BB cyclo D (46, 0.0056%), and E/BC cyclo D (47, 0.0078%), as previously reported [25–29].

Round 2
Reviewer 2 Report
The Authors have complied with the reviewer's suggestion, the manuscript has been revised and may be published in Pharmaceuticals.
Author Response
We are grateful to your reviewing our manuscript and providing valuable suggestions to improve the manuscript. We have incorporated all your comments and suggestions in our revised manuscript. I hope this new manuscript is acceptable for publication in Pharmaceuticals.
Reviewer #2
The Authors have complied with the reviewer's suggestion, the manuscript has been revised and may be published in Pharmaceuticals.
→ Thank you very much for taking the time to review our manuscript.
